# Advancing Renewable Energy Systems: A Numerical Approach to Investigate Nanofluidics’ Role in Engineering Involving Physical Quantities

**DOI:** 10.3390/nano15040261

**Published:** 2025-02-10

**Authors:** Muhammad Abdul Basit, Muhammad Imran, Tayyiba Anwar-Ul-Haq, Chang-Feng Yan, Daniel Breaz, Luminita-Ioana Cotîrlă, Alin Danciu

**Affiliations:** 1Guangzhou Institute of Energy Conversion, Chinese Academy of Sciences, Guangzhou 510640, China; 2School of Energy Science and Engineering, University of Science and Technology of China, Guangzhou 510640, China; 3Department of Mathematics, Government College University Faisalabad, Faisalabad 38000, Pakistan; drmimranchaudhry@gcuf.edu.pk (M.I.);; 4Department of Mathematics and Science Education, Faculty of Education, Biruni University, İstanbul 34015, Turkey; 5Department of Mathematics, “1 Decembrie 1918” University of Alba Iulia, 510009 Alba Iulia, Romania; 6Department of Mathematics, Technical University of Cluj-Napoca, 400114 Cluj-Napoca, Romania; luminita.cotirla@math.utcluj.ro; 7Department of Mathematics, Babes Bolyai University, 400084 Cluj-Napoca, Romania; alin.danciu@ubbcluj.ro

**Keywords:** non-Newtonian/Williamson nanofluid, nanotechnology, stretching wedge, chemical reaction, bioconvection, numerical simulation, heat and mass transfer, renewable energy applications

## Abstract

Nanofluids, with their enhanced thermal properties, provide innovative solutions for improving heat transfer efficiency in renewable energy systems. This study investigates a numerical simulation of bioconvective flow and heat transfer in a Williamson nanofluid over a stretching wedge, incorporating the effects of chemical reactions and hydrogen diffusion. The system also includes motile microorganisms, which induce bioconvection, a phenomenon where microorganisms’ collective motion creates a convective flow that enhances mass and heat transport processes. This mechanism is crucial for improving the distribution of nanoparticles and maintaining the stability of the nanofluid. The unique rheological behavior of Williamson fluid, extensively utilized in hydrometallurgical and chemical processing industries, significantly influences thermal and mass transport characteristics. The governing nonlinear partial differential equations (PDEs), derived from conservation laws and boundary conditions, are converted into dimensionless ordinary differential equations (ODEs) using similarity transformations. MATLAB’s bvp4c solver is employed to numerically analyze these equations. The outcomes highlight the complex interplay between fluid parameters and flow characteristics. An increase in the Williamson nanofluid parameters leads to a reduction in fluid velocity, with solutions observed for the skin friction coefficient. Higher thermophoresis and Williamson nanofluid parameters elevate the fluid temperature, enhancing heat transfer efficiency. Conversely, a larger Schmidt number boosts fluid concentration, while stronger chemical reaction effects reduce it. These results are generated by fixing parametric values as 0.1<ϖ<1.5, 0.1<Nr<3.0, 0.2<Pr<0.5, 0.1<Sc<0.4, and 0.1<Pe<1.5.
This work provides valuable insights into the dynamics of Williamson nanofluids and their potential for thermal management in renewable energy systems. The combined impact of bioconvection, chemical reactions, and advanced rheological properties underscores the suitability of these nanofluids for applications in solar thermal, geothermal, and other energy technologies requiring precise heat and mass transfer control. This paper is also focused on their applications in solar thermal collectors, geothermal systems, and thermal energy storage, highlighting advanced experimental and computational approaches to address key challenges in renewable energy technologies.

## 1. Introduction

A fluid that contains particles as small as a nanometer is referred to as a nanofluid. These fluids cause a base fluid to form colloidal suspensions. In a nanofluid, carbides, oxides, and metals are typically used as the building blocks for the nanoparticles. Nanofluids offer exceptional qualities such as high thermal conductivity, a reduced propensity for clogging within flow conduits, prolonged stability over time, and uniformity because of nanoparticles’ microscopic and incredibly enormous surface areas. Consequently, nanofluids exhibit a diverse array of prospective applications in the domains of electronic cooling systems, pharmacological delivery mechanisms, peristaltic pumping systems for diabetes management, collectors of solar energy, and nuclear technology uses. Choi Eastman [1] introduced the idea of using nanofluids to suspend liquids that include very small particles. Recent investigations have garnered significant interest in the research of non-Newtonian fluids. These substances illuminate and become prominent subjects of inquiry within the domain of fluid mechanics owing to their substantial advantages in the fields of medicine, engineering, mathematics, and various industrial applications. Numerous models have been studied due to the nonlinear connection between stress and deformation rate. Non-Newtonian fluids exhibit numerous practical applications in everyday life, including food preservation, plastic manufacturing, the functionality of diverse lubricants, and the fabrication of paper products. Non-Newtonian fluids include things like cheese, pasta, polymer solutions, ketchup, sugar combinations, syrups, chili sauce, and body lotion. The Williamson fluid model serves as a quintessential illustration of non-Newtonian fluid pseudoplastic fluid properties. This particular fluid experiences an immediate reduction in viscosity concomitant with an increase in shear stress. Williamson has been studied less than other pseudoplastic fluids. Hamied et al. [2] executed an inquiry into the cumulative consequences of an oscillating magnetic field and thermal energy absorption on the steady-state flow dynamics of a non-Newtonian Williamson fluid produced by a stretched cylinder in the presence of nanoparticles. When nano-sized particles were present, Nadeem et al. [3] investigated a Williamson fluid’s two-dimensional flow along a stretched sheet. Another name for this fluid is nano Williamson fluid, and the equations governing the boundary layer for the model are presented and elucidated, in conjunction with the energy component and the volumetric concentration of nanoparticles. In a numerical study, Krishnamurthy et al. [4] examined how radiation and chemical processes affect the uniform flow through the boundary layer produced by a magnetohydrodynamic Williamson fluid through a porous substance in the direction of a horizontally oriented linearly stretching sheet while taking nanoparticles into account. Khan et al. [5] evaluated an analysis of the mathematical ramifications associated with variable viscosity in the context of a Williamson nanofluid situated on a nonlinear stretched surface. Williamson turbulent mixed convective flow with heat transfer with relation to a magnetic field and fluctuating thermal conductivity was presented by Hamid et al. [6]. Maboob et al. [7] undertook an empirical investigation regarding the magnetohydrodynamic flow characteristics of Williamson nanofluid across a continuously advancing thermally heated surface, taking into account an internal heat generation mechanism as well as the effects of thermal radiation. Nayak et al. [8] conducted a comprehensive study regarding the ramifications of Brownian motion and heat exchange in conjunction with chemical interactions affecting the fluid dynamics of Williamson nanofluid as it interacts with a permeable stretching sheet. The potential environmental and economic impacts of integrating Williamson nanofluids into renewable energy systems are significant and merit careful consideration. Environmentally, the enhanced thermal conductivity and improved fluid dynamics of Williamson nanofluids can lead to greater energy efficiency in systems like solar thermal collectors and geothermal energy technologies. This efficiency translates to reduced energy waste and lower greenhouse gas emissions, supporting global efforts to mitigate climate change. However, it is equally important to address the environmental costs associated with the production and eventual disposal of nanofluids, as improper handling of nanoparticles could pose risks to ecosystems. This study’s environmental and economic dimensions underscore the broader implications of adopting advanced nanofluids in renewable energy systems. Future investigations should prioritize life-cycle assessments and techno-economic analyses to further validate these findings and support their practical implementation.

Thermal transport phenomena and electrically conductive fluid dynamics have garnered a lot of attention because of their extensive applicability across various industrial and engineering domains. One of the fundamental mechanisms underlying heat transfer is thermal radiation, which is characterized by the thermal motion of charged particles within the material. Thermal radiation encompasses a diverse array of applications, which may extend beyond, but are not confined to, aerospace technology, thermal regulation factors, modulation of heat transfer processes in polymer manufacturing, phase change materials, solar energy systems, photosynthetic processes, high-capacity cooling solutions, nuclear power generation facilities, space exploration vehicles, and various energy systems. Bhatti et al. [9] demonstrated how heat radiation and thermodiffusion affect Williamson nanofluid when a porous substrate for stretching and shrinking is present. The properties of the dipole with magnetic attraction about shear-thinner Williamson nanofluid with heat radiation effects were thoroughly explored by Khan et al. [10]. The impact of the radiation of heat upon the flow dynamics of Williamson fluid in the setting of an expanding or contracting cylinder that contains nanomaterials was examined by Hamid et al. [11]. A computational evaluation of the irregular (magnetohydrodynamics) flow rate of a Williamson nanofluid in a porous substrate having a heating element or sink and heat radiation concerns was presented by Hamid et al. [12]. A new study was presented by Hashim et al. [13] aimed at creating and comprehending a mathematical framework on the dynamics of a non-Newtonian Williamson fluid in conjunction with nanoparticles while also elucidating the thermal characteristics of the nanoparticles through the utilization of the Rosseland approximation to illustrate the consequence of nonlinear radiative phenomena. Kothandapani et al. [14] described how Williamson nanofluid’s peristaltic motion in an unequal curved channel is affected by the field of magnetic attraction and a heat radiation variable. Rooman et al. [15] performed an investigation pertaining to the optimization of entropy and the examination of heat exchange within the context of MHD Williamson nanofluid flow above a Riga plate that is vertical, taking into account the impacts of thermal radiation that is not linear. Bilal et al. [16] undertook a study about the actions of Williamson magnetic nanofluid over a partially slipping and convective cylinder, incorporating the impacts of infrared radiation and fluctuating heat conductivity. Kumar et al. [17] looked at the impact of Joule heating and the influence of nonlinear heating radiation on the behavior of Williamson nanofluid, emphasizing the phenomenon of entropy generation.

The phrase “activation energy” was first used and introduced by Arrhenius in 1889. According to him, activation energy (AE) is the smallest quantity of energy necessary to start a procedure. The study of fluid dynamics characterized by activation energy has garnered considerable interest from the academic community owing to its substantial applications. The phenomenon of activation energy is integral to numerous processes, including liquid-phase processes, polymerization processes, solid desorption, solid-state decomposition, and electron transfer, and it is extensively utilized in fields such as chemical engineering, geothermal systems, oil emulsions, and food processing. Using binary chemical reaction and heat production, Reddy et al. [18] explored how AE affected the magnetohydrodynamics (MHD) Williamson nanofluid across a wedge that is radiative. Using activation energy (AE) and gyrotactic microorganisms, Aldabesh et al. [19] explored the irregular instantaneous slipping flow properties of Williamson nanofluid. A theoretical study and examination of a Williamson nanofluid’s magnetohydrodynamic (MHD) flow characteristics over an unpredictable extending surface with activation energy considerations was carried out by Dawar et al. [20]. Activation energy’s (AE) influence on mass transport and mixed convection transmission of the heat of a nanofluid created by Williamson with heat production or absorption over a stretched tube was examined by Ibrahim and Negera [21]. Azam et al. [22] proposed a novel concept concerning the optimization of entropy generation alongside the aspects of acoustic emission about the transitory axisymmetric flow of Williamson nanofluid influenced by an axially moving surface and the implications of binary chemical reactions.

The study of structure creation caused by swimming microbe motion is known as bioconvection. Gyrotactic microorganisms often produce density fluctuation and build a thick layer on the liquid’s upper portion. The microorganism descends as the heavy layer splits, causing an upward and downward movement. This phenomenon is identified as bioconvection. In biological systems and biotechnology, bioconvection is used to separate live and dead cells, purify cultures, and create enzyme biosensors. Jabeen et al. [23] performed a theoretical examination of the characteristics of Williamson nanofluid flow through the boundary layer, taking into account various factors such as activation energy AE, viscous dissipation, and bioconvection. Wang et al. [24] conduct a theoretical investigation into the bioconvection process associated with the magnetohydrodynamic flow of Williamson nanofluid on a spreading sheet with different thicknesses while incorporating the consequences of nonuniform viscosity dependent on temperature and heat conductivity. The effects of thermophoretic diffusion and Brownian motion were investigated by Priyadharshini et al. [25], who also took into account the implications of thermal energy and the bioconvection phenomena related to microbes in an asymmetrically extending planar sheet made of a non-Newtonian fluid described by the Williamson model. Patil et al. [26] carried out research into the complex phenomenon of mixed bioconvection within the motion of Williamson nanofluid incorporating the diffusion of liquid oxygen, with the primary contribution being the exploration of bioconvection coupled with roughness of the surface using a different approach through the motion of non-Newtonian Williamson fluid. Basit et al. [27] elucidated the enhancement in heat and mass transfer rates attributable to the presence of motile microorganisms within the flow of Carreau nanofluid through an inclined, stretchable cylindrical structure while considering the effects of activation energy and thermal radiation. By using the second law of thermodynamics, Wu et al. [28] optimized the formation of entropy. Additionally, the tangent hyperbolic nanomaterial model is used to expound on thermophoresis, Brownian diffusions, and slide processes. A mathematical study of the heat and mass transfer efficiency of bioconvective Casson nanofluid flow via a conical gap between rotating surfaces under the effects of activation energy and thermal radiation was conducted by Basit et al. [29]. The efficiency of renewable energy technologies is often constrained by thermal challenges, particularly due to low-temperature heat sources. Nanofluids, which are engineered by dispersing nanoparticles into base fluids, exhibit superior thermal conductivity and heat transfer characteristics, making them highly suitable for renewable energy applications. These unique fluids have shown significant potential in solar thermal systems, geothermal heat pumps, and energy storage technologies, offering a path to optimize energy utilization. Additionally, recent progress in computational modeling and experimental techniques has contributed to a deeper understanding of nanofluid dynamics, enabling their effective integration into decarbonized energy systems.

Renewable energy systems require highly efficient heat and mass transfer mechanisms to meet growing energy demands sustainably. While nanofluids have been extensively studied for their enhanced thermal properties, much of the focus has been on Newtonian fluids, leaving non-Newtonian nanofluids such as Williamson fluids insufficiently explored. Williamson nanofluids, with their shear-thinning characteristics, adapt their viscosity based on flow conditions, which is particularly advantageous in systems with varying operational demands. Despite these benefits, there remains a noticeable gap in the literature regarding the application of Williamson nanofluids in renewable energy technologies. Most existing research focuses on Newtonian nanofluids, overlooking the unique advantages offered by non-Newtonian behavior in complex engineering systems. This study seeks to bridge this gap by investigating the role of Williamson nanofluids in optimizing heat and mass transfer in renewable energy applications, contributing to the development of more efficient and sustainable energy solutions.

Motivated by the above analysis, many researchers considered the numerical exploration of the bioconvective motion of Williamson nanofluids in the vicinity of a stretching wedge, particularly in the context of chemical reaction parameters and the diffusion of liquid hydrogen, but here we also report the practical application of this phenomenon in the renewable energy systems as depicted in Figure 1a–c. Our primary goal is to examine the impacts of activation energy, double diffusion, mixed convection, thermophoresis parameter, chemical reaction parameter, thermal radiation, activation energy, bioconvection, and liquid hydrogen diffusion on the stretching/shrinking wedge. MATLAB bvp4c is used to solve the resulting equations. And also, we present our results in graph and table form. This study unveils the role of nanofluids in overcoming thermal barriers, emphasizing their relevance to modern renewable energy solutions.

## 2. Materials and Methods

The steady laminar and bioconvective nanoliquid motion around a stretching wedge is investigated in this work. It is postulated that the wedge progresses with a velocity denoted as $U_{w}\left(x\right),$ while the free stream’s velocity is designated as $U_{e}\left(x\right)$. It is important to note that the stretching case is shown by $U_{w}\left(x\right)>0,$ whereas a symbol for the shrinking case is $U_{w}\left(x\right)<0$. The Cartesian coordinate system is used to represent the coordinates in which x and y are normal to the surface as well as along it and is used to characterize the flow behavior. u and v stand for the elements of velocity, respectively. The modeled flow geometry configuration is shown in Figure 1d.

Moreover, the surface and mainstream section velocities are specified by $U_{w}$, and $U_{\infty }$, $T_{w}$, $\phi_{w}$ denote the temperature, species diffusion, and volume fraction of nanoparticles in proximity to the wall, whereas $T_{\infty }$, $\phi_{\infty }$ signify these parameters at a distance from the wall, respectively. The governing model of the flow problem is as follows [30]:

Continuity Equation:(1)$$\partial_{x}u+\partial_{y}v=0,$$

Momentum Equation:(2)$$\begin{array}{l} u\partial_{x}u+v\partial_{y}u=U_{e}\left(\partial_{x}U_{e}\right)+\nu \partial_{yy}u+\sqrt{2}\nu \tau \left(\partial_{y}u\right)\left(\partial_{yy}u\right)\\ -\left(\frac{\rho_{p}-\rho }{\rho }\right)g\left(1-\phi_{\infty }\right)\cos\left(\frac{\Omega }{2}\right)+\left(\frac{1}{\rho }\right)\left(1-C_{\infty }\right)g\rho \beta \left(T-T_{\infty }\right)\\ -\left(N-N_{\infty }\right)\gamma g\left(\rho_{m}-\rho \right)-\left(\rho_{p}-\rho \right)g\left(C-C_{\infty }\right), \end{array}$$

Energy Equation:(3)$$u\partial_{x}T+v\partial_{y}T=J\left[D_{B}\partial_{y}\phi \partial_{y}T+\frac{D_{T}}{T_{\infty }}{\left(\partial_{y}T\right)}^{2}\right]+\frac{\nu }{\mathrm{Pr}}\partial_{yy}T,$$

Equation of Conservation of Concentration of Nanoparticles:(4)$$u\partial_{x}C+v\partial_{y}C=D_{B}\partial_{yy}\phi +\frac{D_{T}}{T_{\infty }}\partial_{yy}T+\frac{\nu }{Sc}\partial_{yy}C-k_{r}^{2}{\left(C-C_{\infty }\right)}^{n},$$

Equation of Density of Motile Microorganisms:(5)$$v\partial_{y}n+u\partial_{x}n=Dm\partial_{yy}n-b\frac{W_{c}}{\left(C_{w}-C_{\infty }\right)}\left[n\partial_{y}\left(\partial_{y}C\right)\right]$$

Boundary Conditions:(6)$$\begin{array}{l} y=0:u=U_{w},\;v=0,\;n=n_{w},\;C=C_{w},\;\phi =\phi_{w},\;T=T_{w}\\ y\to \infty :u\to U_{e},\;n\to n_{\infty },\;C\to C_{\infty },\;\phi \to \phi_{\infty },\;T\to T_{\infty } \end{array}$$

Here, *n* is the reaction order, *k* is the reaction rate constant, the volumetric concentration expansion is shown by $\beta_{2}$, the kinematic viscosity is indicated by ν, ρ is the density, Ω represents the angle of inclination of the wedge, and $\beta_{1}$ stands for the volumetric thermal expansion coefficient. The diffusion coefficients for thermophoresis and Brownian motion are shown by the symbols $D_{B}$ and $D_{T}$. μ indicates the fluid viscosity, $D_{n}$ represents the diffusivity of microorganisms, $k_{f}$ denotes the thermal conductivity, $W_{c}$ indicates the maximum speed of the swimming cell, $\rho_{p}$ represents microorganisms’ density, T represents the temperature of the fluid, $T_{f}$ denotes the convective fluid temperature, $T_{\infty }$ is the ambient temperature, n is the density or concentration of microorganisms, $n_{f}$ denotes the surface concentration of microbes, $n_{\infty }$ represents the ambient nanoparticle concentration of microbes, C denotes concentration, $C_{f}$ is taken as the convective fluid concentration, and $C_{\infty }$ denotes the ambient nanoparticle concentration.

The left sides of the equations show the unsteady and convective terms, but on the other side, $-\left(\frac{\rho_{p}-\rho }{\rho }\right)g\left(1-\phi_{\infty }\right)\cos\left(\frac{\Omega }{2}\right)$ dictates buoyancy force due to particle–fluid density differences, modified by porosity and rotation effects; $\left(\frac{1}{\rho }\right)\left(1-C_{\infty }\right)g\rho \beta \left(T-T_{\infty }\right)$ represents a temperature-induced buoyancy effect; $\left(N-N_{\infty }\right)\gamma g\left(\rho_{m}-\rho \right)$ shows the influence of nanoparticle concentration on the buoyancy-driven flow; $\left(\rho_{p}-\rho \right)g\left(C-C_{\infty }\right)$ shows the buoyancy effect due to solutal concentration variations; $\sqrt{2}\nu \tau \left(\partial_{y}u\right)\left(\partial_{yy}u\right)$ involves the kinematic viscosity *ν*, a characteristic time scale *τ*, and velocity gradients in both directions; a *J* coefficient of relates to concentration-driven heat transfer; $\frac{D_{T}}{T_{\infty }}{\left(\partial_{y}T\right)}^{2}$ is a thermophoresis effect; $D_{B}\partial_{y}\phi \partial_{y}T$ is Brownian motion; $\frac{\nu }{Sc}$ is mass diffusion; and $k_{r}^{2}{\left(C-C_{\infty }\right)}^{n}$ is the reaction kinetics.

Similarity Transformation:(7)$$\begin{array}{l} \psi ={\left(\frac{2x\nu U_{e}}{\left(m+1\right)}\right)}^{\frac{1}{2}}f\left(\eta \right),\;\eta ={\left(\frac{\left(m+1\right)U_{e}}{2x\nu }\right)}^{\frac{1}{2}}y,\;\theta =\left(T-T_{\infty }\right)\frac{1}{\left(T_{w}-T_{\infty }\right)},\\ \phi =\left(C-C_{\infty }\right)\frac{1}{\left(C_{w}-C_{\infty }\right)},\;\;\chi =\left(n-n_{\infty }\right)\frac{1}{\left(n_{w}-n_{\infty }\right)} \end{array}$$

Here, ψ shows the role of the stream function, identified as $u=\partial_{y}\psi$ and $v=-\partial_{x}\psi$, which fulfills the continuity Equation (1). Consequently, this leads to(8)$$u=f^{\prime }\left(\eta \right)U_{e},v=-{\left(\frac{\left(m+1\right)U_{e}\nu }{2x}\right)}^{\frac{1}{2}}\left(f\left(\eta \right)+\frac{m-1}{m+1}\eta f^{\prime }\left(\eta \right)\right),$$

Considering Equation (7), the reduction of Equations (2)–(6) are as follows:(9)$$\begin{array}{l} f^{\prime \prime \prime }\left(\eta \right)\left(1+Wf^{\prime \prime }\left(\eta \right)\right)+\frac{2m}{m+1}\left(1-{\left(f^{\prime }\left(\eta \right)\right)}^{2}\right)+\frac{2\times Ri}{1+m}\cos\left(\frac{\Omega }{2}\right)+\\ f\left(\eta \right)f^{\prime \prime }\left(\eta \right)+\omega \left(\theta -Nr\phi -Rb\chi \right)=0 \end{array}$$(10)$$\theta^{\prime \prime }\left(\eta \right)+\mathrm{Pr}\left(1+m\right)f\left(\eta \right)\theta^{\prime }\left(\eta \right)+\mathrm{Pr}Nb\phi^{\prime }\left(\eta \right)\theta^{\prime }\left(\eta \right)+\mathrm{Pr}Nt{\left(\theta^{\prime }\left(\eta \right)\right)}^{2}=0$$(11)$$\phi^{\prime \prime }\left(\eta \right)+Sc*Lef\left(\eta \right)\phi^{\prime }\left(\eta \right)-\frac{2}{m+1}\times Kc+\frac{Nt}{Nb}\theta^{\prime \prime }\left(\eta \right)=0$$(12)$$\chi^{\prime \prime }-\frac{2}{m+1}Lb\left(f\chi^{\prime }\right)-Pe\left(\phi^{\prime \prime }\left(\chi -\delta_{1}\right)+\chi^{\prime }\phi^{\prime }\right)=0$$

The following dimensionless traits are used in the analysis:Ri=GrRe2,  Nt=JDTTw−T∞T∞ν,  Le=νDB,  J=ρpCppρfCpf,  Nr=ρp−ρθW−θ∞ρβ1TW−T∞1−θ∞,Nb=JDBθW−θ∞ν,Kc=k1C−C∞n−1xue,  Wi=τ2m+1ue3νx12,  Nc=β3CW−C∞β1TW−T∞,Rb=ρm−ρNw−N∞γρβ1−C∞Tw−T∞,  ω=1−C∞βgTw−T∞c12r,

The following is skin friction:(13)Re12Cf=τwρue2=1+m2f″0+We2×f″02

The strength of energy transfer is as follows:(14)Re−12Nu=xqwkTw−T∞=−θ′0×1+m2

Here is the mass transport strength:(15)Re−12Sh=xJwDBCw−C∞=−ϕ′0×1+m2

The following is the nanoparticle mass transport strength:(16)Re−12Nun=xpwDBCw−C∞=−χ′0×1+m2

Here,(17)$$\tau_{w}=\mu \left(\partial_{y}u+\frac{\tau }{\sqrt{2}}{\left(\partial_{y}u\right)}^{2}\right),\;q_{w}=-k\partial_{y}T,\;J_{w}=-D_{B}\partial_{y}C,\;p_{w}=-k\partial_{y}S\;\mathrm{at}\;y=0$$

The dimensionless boundary conditions are(18)$$\mathrm{At}\;\eta =0:\;\;\;f^{\prime }\left(\eta \right)=\lambda ,\;\;\;\theta =1,\;\;\;\phi =1,\;\;\;\chi =1,\;\mathrm{As}\;\eta \to \infty :\;\;\;f^{\prime }\left(\eta \right)=1,\;\;\;\theta =0,\;\;\;\phi =0,\;\;\;\chi =0.$$

## 3. Numerical Scheme and Computational Approach

The modified Equations (9)–(12), along with their respective boundary conditions, are employed computationally using a shooting approach with the bvp4c solver in MATLAB (https://www.mathworks.com/products/matlab.html, accessed on 6 February 2025). The steps to be adopted for the implementation of the numerical scheme are depicted in the flow chart in Figure 1e. The limitation of this scheme is that it will give more refined results as it has an error bound 10^−6^.

In the following segment, the higher-ordered nonlinear ordinary differential equations (ODEs) are converted into first-order ODEs by taking into account the following variables:$$\begin{array}{l} f=s_{1},\;f^{\prime }=s_{2},\;f^{\prime \prime }=s_{3},\;f^{\prime \prime \prime }=s_{3}^{\prime },\\ \theta =s_{4},\;\theta^{\prime }=s_{5},\;\theta^{\prime \prime }=s_{5}^{\prime },\\ \phi =s_{6},\;\phi^{\prime }=s_{7},\;\phi^{\prime \prime }=s_{7}^{\prime },\\ \chi =s_{8},\;\chi^{\prime }=s_{9},\;\chi^{\prime \prime }=s_{9}^{\prime }. \end{array}$$

Then we have(19)$$s_{3}^{\prime }=\frac{1}{\left(1+Ws_{3}\right)}\left[\begin{array}{l} -\frac{2m}{m+1}\left(1-s_{2}^{2}\right)\\ -\frac{2\times Ri}{1+m}\left(s_{4}+Nc\times s_{6}-Nr\times s_{6}\right)\cos\left(\frac{\Omega }{2}\right)\\ +s_{1}s_{3}+\omega \left(s_{4}-Nrs_{6}-Rbs_{8}\right) \end{array}\right]$$(20)$$s_{5}^{\prime }=-\mathrm{Pr}\left(1+m\right)s_{1}s_{5}-\mathrm{Pr}Nbs_{7}s_{5}-\mathrm{Pr}Nts_{5}^{2}$$(21)$$s_{7}^{\prime }=-Sc^{*}Les_{1}s_{7}+\frac{2}{m+1}\times Kc-\frac{Nt}{Nb}s_{5}^{\prime }$$(22)$$s_{9}^{\prime }=\frac{2}{m+1}Lbs_{1}s_{9}+Pe\left(s_{7}^{\prime }\left(s_{8}-\delta_{1}\right)+s_{9}s_{7}\right)$$

## 4. Discussion of Results

In this section, a complete discussion about the results achieved from our developed mathematical model of nanofluid flow throughout a wedge involving chemical reactions and motile microorganisms is presented. Figure 2 displays the effect of the Williamson parameter ϖ on the velocity parameter. As the dimensionless parameter ϖ increases, the fluid motion also increases. Figure 3 illustrates how the Falkner–Skan power law parameter m affects the velocity field parameter. It demonstrates how the flow’s velocity appears to rise noticeably as the pressure gradient increases $\left(m>0\right)$. This observation suggests that the nanofluid flow experiences significant acceleration. Before separation occurs, the laminar boundary layer may tolerate very slight blockage. Figure 4 shows the fluctuation when the buoyancy ratio parameter Nr increases and how it affects the velocity field $f^{\prime }\left(\eta \right)$. The buoyancy ratio parameter Nr increases both the velocity and the thickness of the thermal boundary. Energy transmission of the fluid is the cause of this increase. Figure 5 illustrates how the velocity parameter is affected by the bioconvection Rayleigh number $\left(Rb\right)$. An elevation in the ratio of Rb corresponds to an enhancement in the velocity parameter. This parameter exhibits an inverse relationship to $\left(N_{w}-N_{\infty }\right)$, and buoyancy effects are minimized to demonstrate the flow. The bioconvection Rayleigh number is a dimensionless quantity that captures the combined influence of buoyancy and viscous forces on fluid flow in the presence of microorganisms. While commonly used in biological contexts, this parameter has applications in geothermal energy systems. In these systems, buoyancy-driven convection plays a vital role in improving heat transfer within porous media. A higher Rayleigh number indicates a stronger buoyancy effect, which can potentially enhance the efficiency of heat extraction from geothermal wells. Figure 6 depicts how the velocity parameter $f^{\prime }\left(\eta \right)$ is affected by the Williamson variable W. As the Williamson variable increases, the velocity gradient becomes more pronounced. Enhancements in the resistance that become more effective as the sharing rate increases are caused by an increase in the rate at which velocity varies. When the velocity distribution $f^{\prime }\left(\eta \right)$ increases due to high values of the Williamson fluid parameter, this impact becomes increasingly visible.

Figure 7 shows how the Williamson nanofluid’s Brownian motion Nb effects the temperature profile $\theta \left(\eta \right)$. Physically, the surrounding particles are moved from hot to cold zones by the Brownian motion forces. Therefore, a higher temperature results from an enhanced Brownian motion input.

Figure 8 illustrates how the thermophoretic parameter Nt effects the dimensionless temperature parameter $\theta \left(\eta \right)$. As the thermophoretic parameter got higher, we found that the temperature profile also increased. The temperature gradient of thermophoretic forces, which includes high-velocity flow away from the surface, is what causes this profile rise. In this manner, hot fluid escapes the surface, and as a result, Nt rises. In renewable energy applications like solar thermal systems, thermophoresis influences the distribution and stability of nanoparticles within nanofluids. This effect is crucial because it can contribute to better heat absorption in solar collectors and mitigate issues like nanoparticle aggregation or sedimentation over time. Thermophoretic effects help maintain the uniform dispersion of nanoparticles, thereby improving the overall thermal performance and longevity of the fluid in renewable energy systems. Figure 9 illustrates how the temperature profile $\theta \left(\eta \right)$ falls as the Prandtl number Pr increases. The dimensionless Prandtl number serves as a metric for quantifying the ratio of thermal diffusivity to mass diffusivity. Accordingly, a reduction in the Prandtl number signifies an increase in thermal diffusivity relative to mass diffusivity. The thermal field decreases as a result of this occurrence.

The influence exerted by the chemical reaction parameter $\left(Kc\right)$ on the concentration profile $\phi \left(\eta \right)$ is seen in Figure 10. There is a decline in the fluid’s concentration as the parameter of chemical reaction rises. Figure 11 illustrates how the concentration profile $\phi \left(\eta \right)$ and the Lewis parameter Le are related. The graph’s input illustrates how the concentration profile deteriorates as the Lewis number values rise. It illustrates how the Lewis parameter lowers the concentration diffusivity coefficient of nanoparticles. The impact of the thermophoresis parameter Nt on the concentration profile $\phi \left(\eta \right)$ is seen in Figure 12. With rising Nt values, the concentration of nanoparticles increases as well. Physically, the liquid’s particles move from the heated to the cool section, raising the concentration of nanoparticles. The characteristics and features of Sc are presented in Figure 13. In this investigation, raising the Schmidt parameter values resulted in a drop in the concentration profile $\phi \left(\eta \right)$. In actuality, the Schmidt number is calculated by dividing the momentum diffusivity by the temperature diffusivity. A large change in Sc causes the momentum diffusivity to increase and, thus, the concentration profile to decrease.

The association between the temperature difference parameter δ and the motile microbe is well illustrated in Figure 14. It is clear that the microorganism profile rises as the temperature difference parameter rises. It is clear from Figure 15 that the density of the motile microbe $\chi \left(\eta \right)$ rises with rising bioconvection Lewis number Lb values, which are proportional to the mass diffusivity of microbes. The differences in the motile microbe profile $\chi \left(\eta \right)$ over a range of Peclet number Pe values are seen in Figure 16. It demonstrates that as the Peclet number grows, so the profile of motile microorganisms also increases. A high value of the Peclet parameter indicates a considerable direction of microorganism movement, which raises the microorganism profile. The comparative power of focused and random swimming motile microorganisms is physically assessed by the Peclet number.

The findings of this study carry significant implications for practical applications in renewable energy systems. The enhanced heat transfer capabilities observed with Williamson nanofluids can play a crucial role in improving the efficiency of solar thermal collectors by facilitating better energy absorption and distribution. This study’s numerical analysis of parameters, such as the Hartmann number and thermal stratification, provides insights into how magnetic fields and temperature gradients affect fluid behavior. These insights are particularly valuable for developing magnetic cooling technologies and optimizing porous media flow in hydrogen fuel cells.

### Validation Table of Obtained Results

We compared our obtained results with the available literature references and found a great accord with them, which validates our claim (Table 1).

## 5. Conclusions

The present research investigates the influence of activation energy, Brownian motion, bioconvection, chemical reactions, and thermophoresis on the Williamson nanofluid flow across a wedge. Applying suitable similarity transformations, a collection of governing equations were simplified to ordinary differential equations (ODEs), and these were then subsequently resolved numerically using the bvp4c module in MATLAB and using shooting method. In the end, the behavior of the velocity $f^{\prime }\left(\eta \right)$, temperature $\theta \left(\eta \right)$, concentration $\phi \left(\eta \right)$, and motile microorganism $\chi \left(\eta \right)$ fields were examined graphically. The main conclusions of the present investigation are listed below:➢Increasing the bioconvection parameter increases liquid velocity.➢The resulting velocity field grows by raising the values of ϖ, m, Nr, Rb, and W.➢The liquid’s temperature rises as the Brownian motion Nb and thermophoretic parameter Nt values increase.➢Furthermore, the temperature profile falls as the Prandtl number rises.➢The concentration profile decreases by increasing the values of Kc, Le, Sc.➢$\phi \left(\eta \right)$ increases by raising the value of the thermophoresis parameter Nt.➢The field of motile microorganisms $\chi \left(\eta \right)$ is increased by enhancements in the temperature differential parameter, bioconvection Lewis number, and Peclet number.

From the outcomes, it is observed that nanofluids have superior thermal properties, and by integrating advanced modeling techniques, nanofluids pave the way for sustainable and efficient energy solutions, contributing significantly to decarbonization and the future of renewable energy. Key findings show that Williamson nanofluids offer substantial improvements in energy efficiency and system performance, making them a promising candidate for future renewable energy technologies.

Future research should focus on exploring the effects of additional parameters, such as particle shape and concentration, to enhance the performance of renewable energy systems. This work contributes to the growing body of knowledge on nanofluids and sets the stage for their integration into more efficient and sustainable energy solutions.

## Figures and Tables

**Figure 1 nanomaterials-15-00261-f001:**
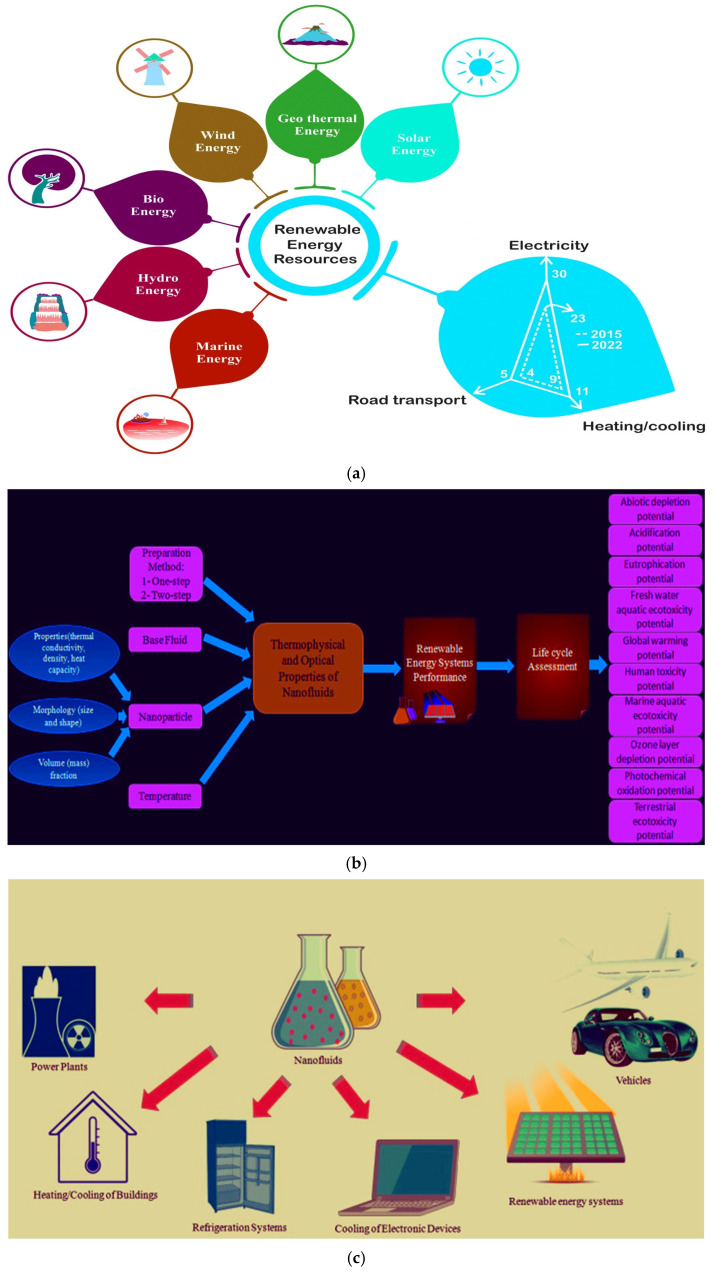
(**a**) Renewable energy applications. (**b**) Development of an energy system from nanoparticles. (**c**) Applications of nanofluids. (**d**) Flow geometry of the flow problem. (**e**) Flowchart of the numerical scheme.

**Figure 2 nanomaterials-15-00261-f002:**
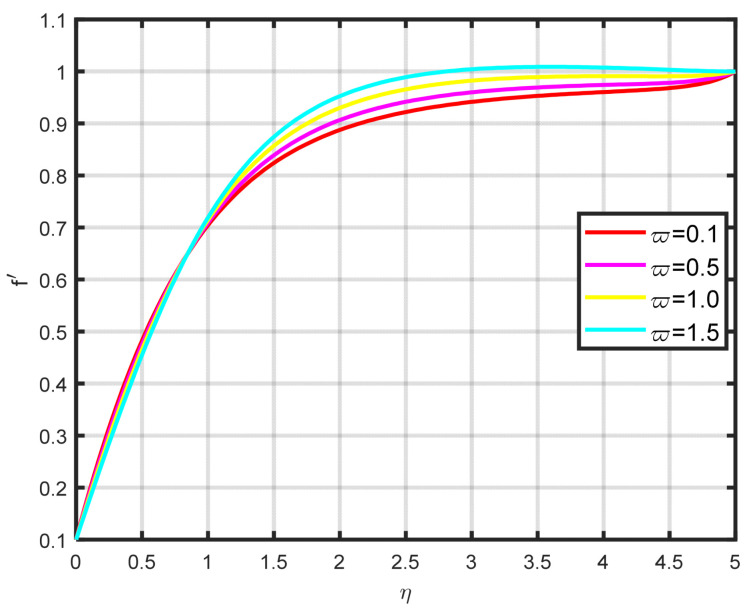
The impact of $f^{\prime }\left(\eta \right)\;\mathrm{on}\;\varpi$.

**Figure 3 nanomaterials-15-00261-f003:**
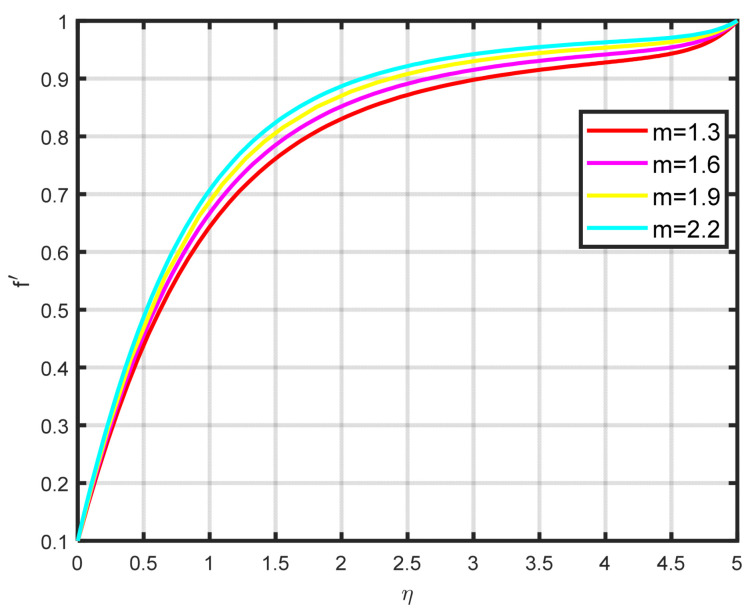
The effects of $f^{\prime }\left(\eta \right)\;\mathrm{on}\;m$.

**Figure 4 nanomaterials-15-00261-f004:**
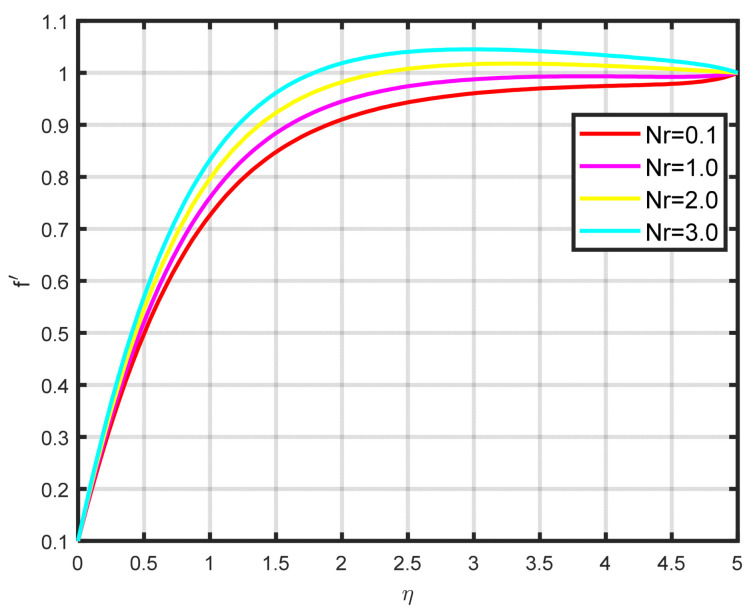
The impact of $f^{\prime }\left(\eta \right)\;\mathrm{on}\;\mathrm{Nr}$.

**Figure 5 nanomaterials-15-00261-f005:**
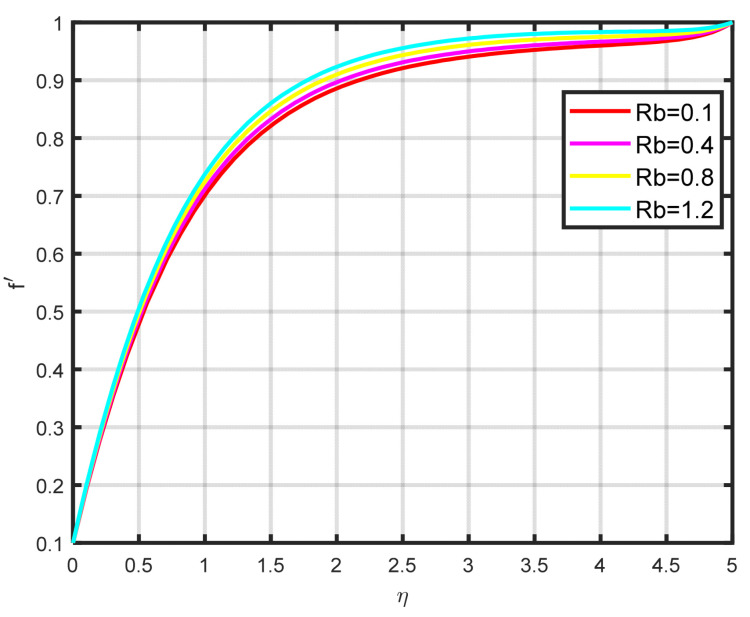
The impact of $f^{\prime }\left(\eta \right)\;\mathrm{on}\;\mathrm{Rb}$.

**Figure 6 nanomaterials-15-00261-f006:**
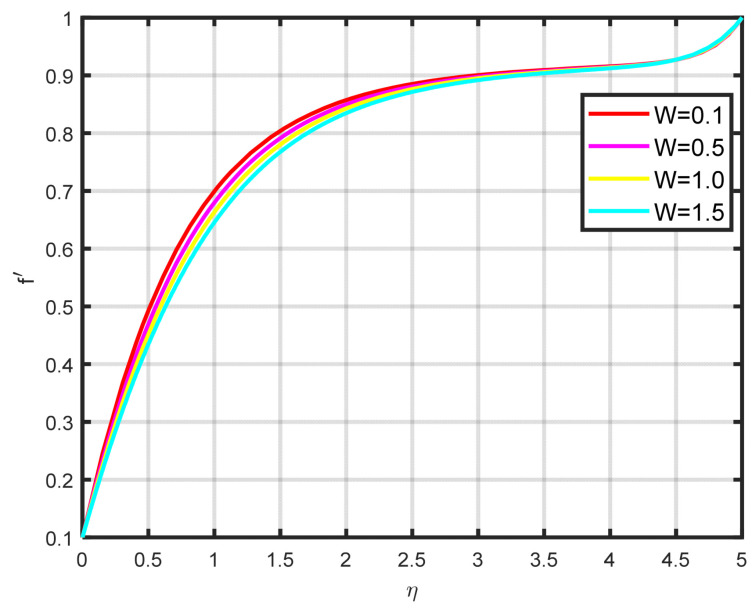
The impact of $f^{\prime }\left(\eta \right)\;\mathrm{on}\;W$.

**Figure 7 nanomaterials-15-00261-f007:**
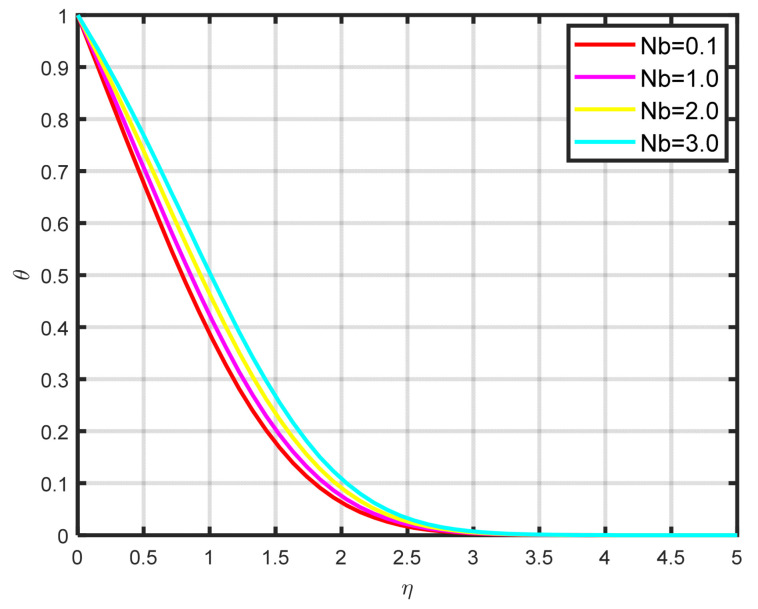
The impact of $\theta \left(\eta \right)$ on Nb.

**Figure 8 nanomaterials-15-00261-f008:**
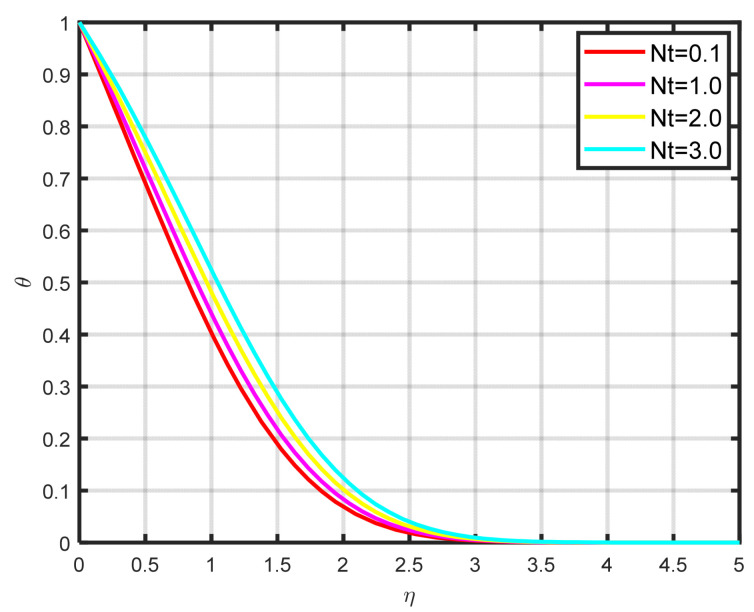
The impact of $\theta \left(\eta \right)$ on Nt.

**Figure 9 nanomaterials-15-00261-f009:**
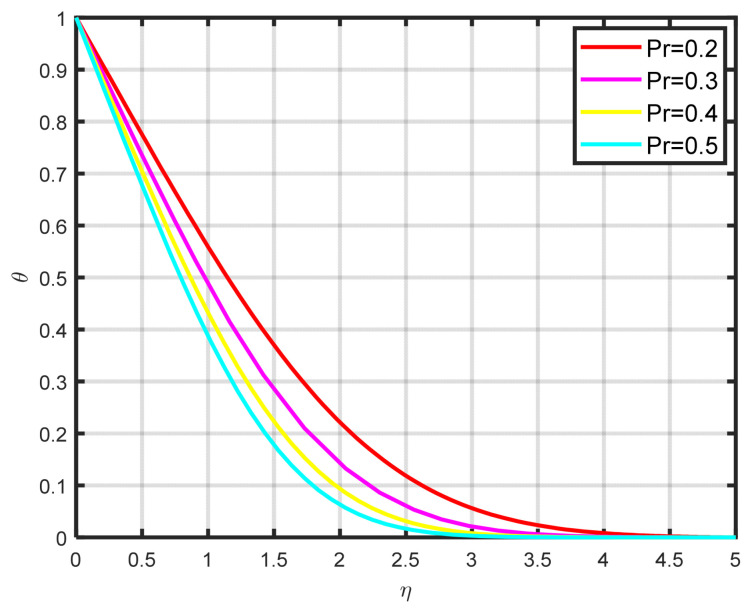
The impact of $\theta \left(\eta \right)\;\mathrm{on}\;\mathrm{Pr}$.

**Figure 10 nanomaterials-15-00261-f010:**
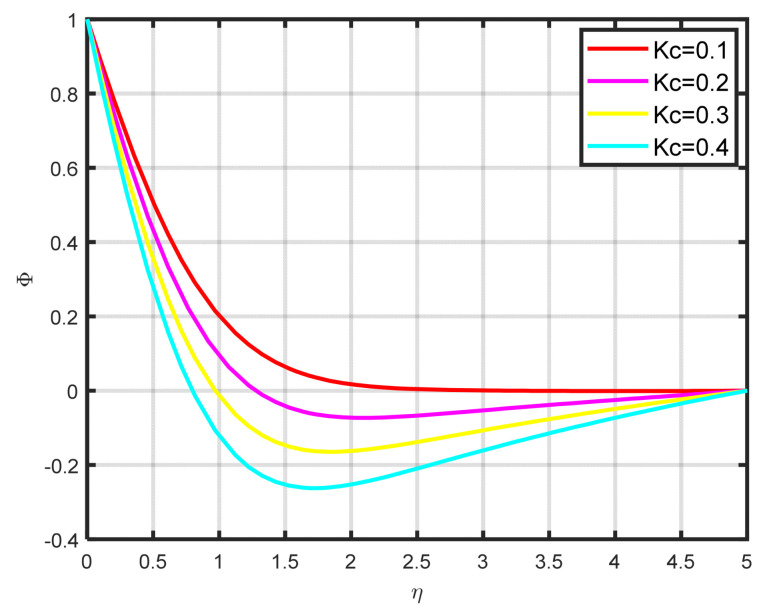
The impact of $\phi \left(\eta \right)\;\mathrm{on}\;\mathrm{Kc}$.

**Figure 11 nanomaterials-15-00261-f011:**
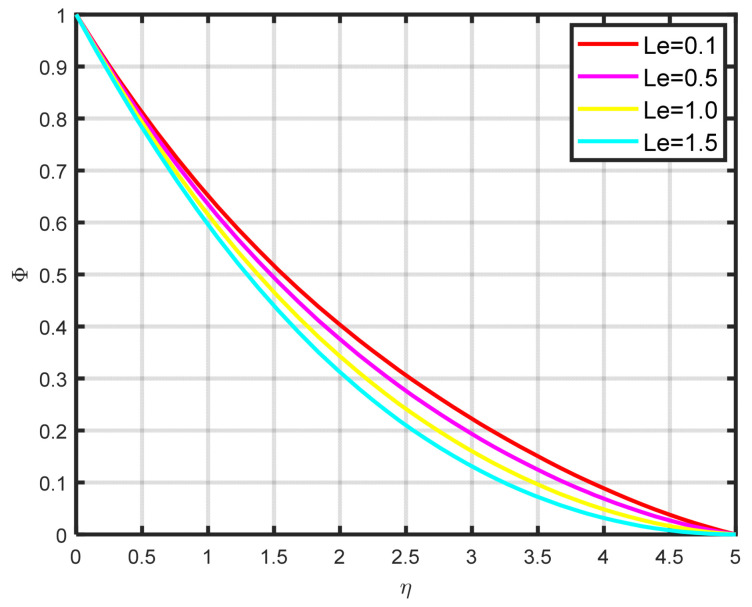
The impact of $\phi \left(\eta \right)\;\mathrm{on}\;\mathrm{Le}$.

**Figure 12 nanomaterials-15-00261-f012:**
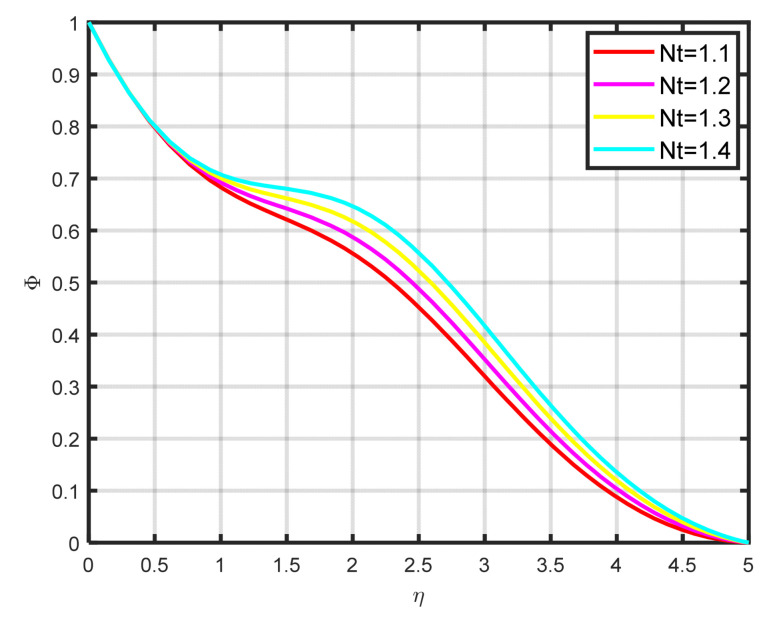
The impact of $\phi \left(\eta \right)\;\mathrm{on}\;\mathrm{Nt}$.

**Figure 13 nanomaterials-15-00261-f013:**
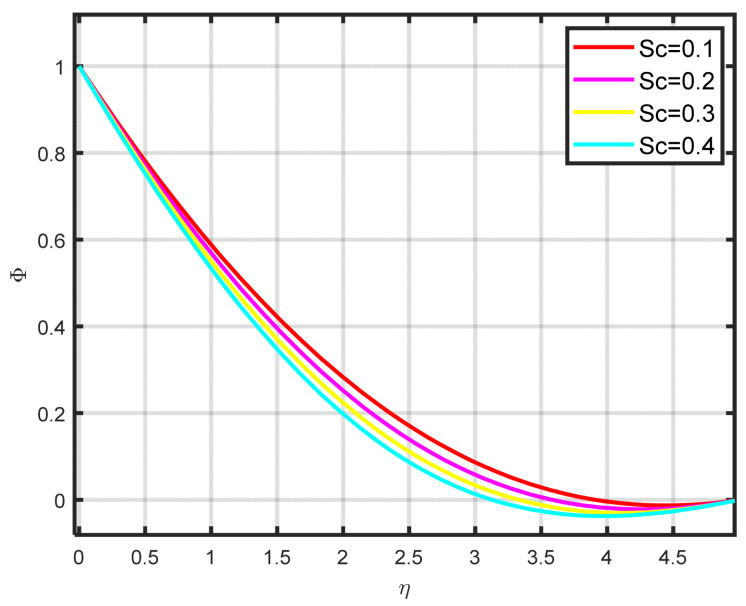
The impact of $\phi \left(\eta \right)\;\mathrm{on}\;\mathrm{Sc}$.

**Figure 14 nanomaterials-15-00261-f014:**
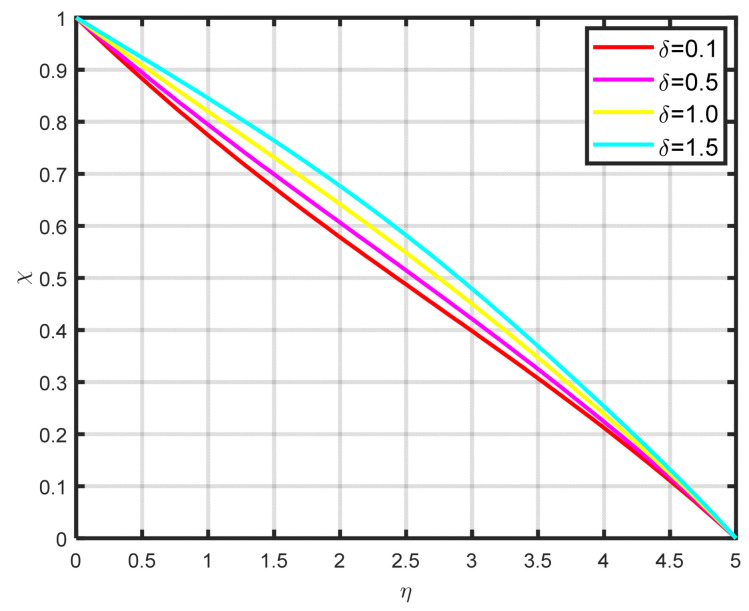
The impact of δ.

**Figure 15 nanomaterials-15-00261-f015:**
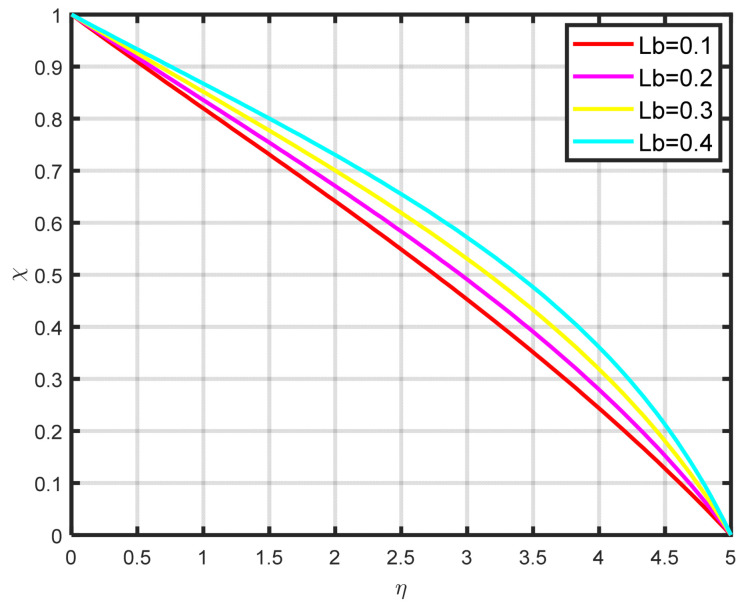
The impact of $\chi \left(\eta \right)\;\mathrm{on}\;\mathrm{Lb}$.

**Figure 16 nanomaterials-15-00261-f016:**
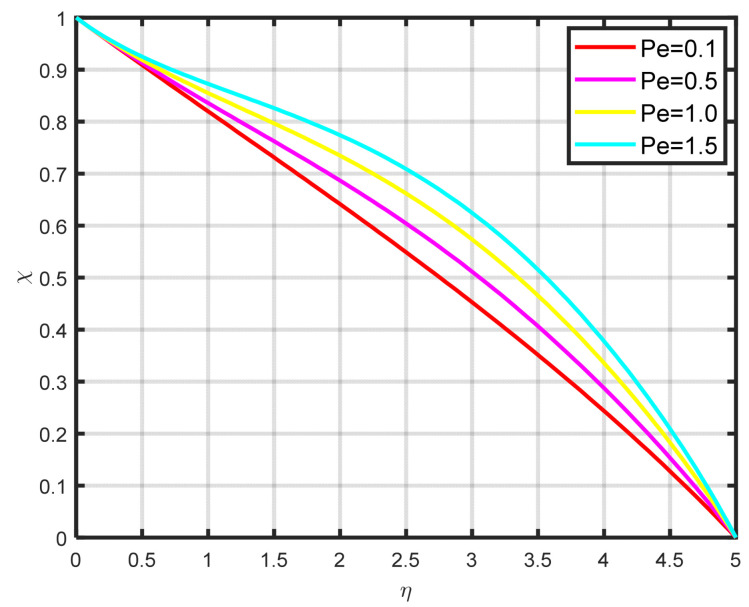
The impact of $\chi \left(\eta \right)\;\mathrm{on}\;\mathrm{Pe}$.

**Table 1 nanomaterials-15-00261-t001:** The numeric values of $f^{\prime }\left(\eta \right)$ by varying the values of the stretching parameter.

Stretching Parameter	Waini et al. [31]	Kulkarni and Shankar [30]	Present Results
0.0	1.232588	1.232520	1.232521
0.5	1.541751	1.541738	1.541740
1.0	1.889331	1.889330	1.889330

## Data Availability

Data will be made available on a reasonable request to the corresponding author.

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
