# Peer review of "Advancing Renewable Energy Systems: A Numerical Approach to Investigate Nanofluidics’ Role in Engineering Involving Physical Quantities"

_nanomaterials, 2025, doi:10.3390/nano15040261_

Round 1
Reviewer 1 Report
Comments and Suggestions for Authors
nanomaterials-3402848:
*Paper lacks a strong argument for the importance of using Williamson nanofluids in the specific renewable energy applications explored. A more robust justification based on existing gaps in the literature is needed.
*While the study presents a numerical approach, it does not suficiently connect the theoretical findings to practical applications, particularly in renewable energy systems.
*Although numerical results are discussed, there is no mention of experimental validation or comparison with experimental data, which would strengthen the findings.
*Some parameters, such as the bioconvection Rayleigh number and thermophoretic effects, are not explained adequately in terms of their physical significance in the context of renewable energy.
*Many figures lack detailed captions and units on axes, making them difficult to interpret without additional context.
*Manuscript is dense with technical terms without sufficient explanation, which could limit acessibility for a broader audience.
*There is no discussion of uncertainties or potential limitations in the numerical methods used, such as the impact of assumptions on results.
*While the article references several studies, it does not clearly highlight how its findings compare to or advance beyond existing work.
*Abstract fails to concisely summarize the novelty and primary findings of the study, making it less engaging for readers.
*Conclusions are repetitive and do not provide specific, actionable insights or highligt the study's unique contributions.
*Although renewable energy is a central theme, the article does not assess the potential environmental or economic impacts of integrating Williamson nanofluids into such systems.
*Some symbols in the equations and nomenclature are not defined consistently throughout the text, causing confusion.
*Manuscript lacks a forward-looking perspective, such as suggestions for experimental studies or real-world application testing.
*Methodology for solving the governing equations is described briefly and would benefit from a more detailed explanation or a flowchart summarizing the steps.
*Text contains several grammatical errors and awkward phrasing, which detract from the clarity and professionalism of the manuscript.
**After these corrections, a new detailed review will be conducted.
Comments on the Quality of English LanguageThe English could be improved to more clearly express the research.
Author Response
Dear Reviewer,
I hope you are doing well. We have addressed your kind comments as per guidelines. The detailed response to your Comments are attached with this reply.

Reviewer 2 Report
Comments and Suggestions for Authors
There is a huge contrast in the article. There is a six-page introduction reduced to the level of scientific dissemination, followed by a three-page collection of formulas, where even minimal explanations and interpretations are missing.
However, this should be described in much more detail. What do we see in Figure 1e at all? What are the blue stars and yellow dots? It would be simpler to represent the velocity field separately with vectors, the temperature field with colors, etc. There is a downward arrow for gravity. But what is the other similar one also pointing upwards?
What is the physical content of the equations? Only the continuity equation can be considered trivial. What is the physical interpretation of the individual terms and their applicability in mathematical expressions?
Chapter 2 should follow the structure and style of the scientific results presentation. In its present form, it is a collection of formulas, without any explanation, involving their validity, the time dependence or independence of the problem, etc.
Moreover, this section does not contain a single reference. Does it mean that the authors constructed all of the equations?
The figures provided in the discussion are probably good, but their interpretation largely consists of showing how the curves will be distorted if the given parameter changes. At the same time, in the spirit of the introduction, much clearer and easier-to-understand explanations and real-world examples with measurement data should be given to the reader.
p. 6, line 184: The authors write: “…renewable energy systems as depicted in Figure 1(a-d).” - The explanations for the figures should be provided earlier. For example, Figure 1a should be referenced on page 5, line 169 (or thereabouts). On the other hand, I do not understand the role of Figure 1b. I think it can be omitted.
In Section 4, figures should be inserted into the text if possible.
Please, check the text. Misprints (e.g.):
p. 2, line 70: nono-sized - nano-sized
p. 3, line 120: magnto – magneto
p. 3, line 122: joule – Joule
Author Response

(The authors gave the same response as above.)

Round 2
Reviewer 1 Report
Comments and Suggestions for Authors
nanomaterials-3402848R1:
Good review done.
Comments on the Quality of English LanguageThe English could be improved to more clearly express the research.
Author Response
Dear Reviewer,
Thank you very much for the valuable comment "Good review done" and providing your critical comments for the improvement of our manuscript.
We have proof read the manuscript again on your recommendation for English improvement purpose and make every sentence clear/understandable.
Best Regards,
Authors of the manuscript
Reviewer 2 Report
Comments and Suggestions for Authors
The article is much better in its current form. However, I would like to return to one of my earlier criticisms:
“Chapter 2 should follow the structure and style of the scientific results presentation. In its present form, it is a collection of formulas, without any explanation, involving their validity, the time dependence or independence of the problem, etc.”
The Authors reply is: “… we have described every equation and involving terms properly in
main text those are needed and remaining in the nomenclature. Please see the revised manuscript.”
I think the criticism was misunderstood. In a physics paper, the physical meaning of the terms in the equations must be described in text. It is not enough to say "here are these equations".
Only the continuity equation can be considered trivial. Of course, here too it should be added that in the stationary case. As in the other cases.
But equations 2-5 contain expressions that are not trivial at all. These must be written out in text, that the nth term expresses/gives/formulates the effect of ..., etc. After the equation, for each term. And the physical meaning must be listed with the validity: e.g., as long as the temperature difference is small, as long as the effect of pressure is negligible, the viscosity is …., etc.
These must be described in detail here.
The later equations can be left to the reader, although it is worth helping them understand them.
Author Response
Dear Reviewer,
Thank you very much for providing your critical comments for the improvement of our manuscript. The detailed response to your comment is attached.
Best Regards,
Authors of the manuscript

Round 3
Reviewer 2 Report
Comments and Suggestions for Authors
please refer to the attachment
